# Nano-hydroxyapatite improves intestinal absorption of acetazolamide (BCS Class IV drug)–but how?

**Kenichi Kaneko** [ORCID] *[⊕], **Ryosuke Miyasaka**[⊕], **Roslyn Hayman**

Sangi Co., Ltd, Central Research Laboratory, Kasukabe, Saitama, Japan

[⊕] These authors contributed equally to this work.

* kaneko@sangi-j.co.jp

**Data Availability Statement:** All relevant data are within the paper and its Supporting Information files.

## Abstract

We earlier reported that coating poorly water-soluble drugs with nano-hydroxyapatite (nano-HAP) improves bioavailability after oral administration. In the present study, we coated BCS Class IV drug acetazolamide (AZ) with nano-HAP (AZ/HAP formulation), and investigated its bioavailability and nano-HAP's role in promoting it. *We tested AZ bioavailability after a single oral dose of the AZ/HAP formulation in rats*, followed by a series of *in vitro*, *ex vivo* and *in vivo* testing. The binding state of AZ and nano-HAP was analyzed by gel filtration chromatography. AZ permeability was studied using a Caco-2 cell monolayer assay kit, to test for tight junction penetration, then using an Ussing chamber mounted with intestinal epithelium, both with and without Peyer's patch tissue, to examine the role of intracellular transport. Fluorescence-labeled nano-HAP particles were administered orally in rats to investigate their localization in the intestinal tract. The area under the blood concentration time-curve in rats was about 4 times higher in the AZ/HAP formulation group than in the untreated AZ group. Gel filtration analysis showed AZ and nano-HAP were not bound. The Caco-2 study showed equivalent AZ permeability for both groups, but without significant change in transepithelial electrical resistance (TEER), indicating that tight junctions were not penetrated. In the Ussing chamber study, no significant difference in AZ permeability between the two groups was observed for epithelium containing Peyer's patch tissue, but for epithelium without Peyer's patch tissue, at high concentration, significantly higher permeability in the AZ/HAP formulation group was observed. Fluorescent labeling showed nano-HAP particles were present in both intestinal villi and Peyer's patch tissue 30 min after oral administration. Our results suggest that nano-HAP's enhancement of drug permeability from the small intestine occurs not via tight junctions, but intracellularly, via the intestinal villi. Further study to elucidate the mechanism of this permeability enhancement is required.

## Introduction

In recent years, new drug development has become increasingly difficult and more expensive worldwide. It has been reported that the average success rate for new drug development is no

**Funding:** This research was funded by Sangi Co. Ltd, to which the authors belong, and received no specific grant from any funding agency. The funder provided support in the form of salaries for authors KK and RM, but did not have any additional role in the study design, data collection and analysis, decision to publish, or preparation of the manuscript. The specific roles of the authors are articulated in the 'author contributions' section.

**Competing interests:** I have read the journal's policy and the authors of this manuscript have the following competing interests: Kenichi Kaneko and Ryosuke Miyasaka are employees of Sangi Co., Ltd. Roslyn Hayman is the president of Sangi Co., Ltd. This does not alter the author's adherence to all the PLOS ONE policies on sharing data and materials, in this study.

greater than about 2% [1]. Long development times, a significant decrease in the approval of new drugs and the expiration of patents have led to a decline in profits for pharmaceutical companies, threatening the sustainability of the industry.

In the past, new drug candidates have been sought by high-throughput screening of synthetic libraries and *in silico* research to identify possible drugs from the structures of target molecules. These computer technologies have since evolved into artificial intelligence (AI) systems, based on huge amounts of data, which are now used for drug repositioning [2,3]. Yet the number of approved new drugs is decreasing, despite an increase in development pipelines worldwide [4]. One reason is that, in the drug development process described above, many target-optimized drug candidates tend to be poorly water-soluble, which results in poor bioavailability when taken orally [5]. Of the various ways of administering a drug, from the point of view of patient quality of life (QOL), oral administration is most desirable. Yet almost 40% of marketed drugs and 70% of candidate compounds were classified as poorly water-soluble at the beginning of the 2000s [6,7]. Drugs are classified into four types by the U.S. Biopharmaceutics Drug Classification System (BCS) according to their solubility in water and permeability across biological membranes [8,9]. Drugs in BCS Classes II (low solubility, high permeability), III (high solubility, low permeability) and IV (low solubility, low permeability) show low bioavailability when orally administered. Therefore, in the process of formulation, efforts have been made to improve bioavailability by various techniques such as particle size reduction, lipid-based formulations, solid dispersions, and absorption enhancers [10–14].

We previously reported successfully improving the solubility of a number of poorly water-soluble drugs by mechanically coating the drug surface with nano-hydroxyapatite (nano-HAP), which is the main component of mammalian teeth and bone and a highly biocompatible substance [15]. A feature of this method is that no physical or chemical pretreatment of the drug is necessary.

In the case of bezafibrate (BCS Class II), we found that the area under the blood concentration-time curve (AUC) after oral administration in rats increased when using the nano-HAP coated formulation compared with bezafibrate alone. Moreover, it was found that the side effects observed on continuous administration of bezafibrate were reduced when using the nano-HAP coating [15]. In the case of cisplatin, a highly polar broad spectrum anti-tumor drug classified under BCS Classes III and IV and administered intravenously, lowering patient QOL [16,17], we found that coating with nano-HAP improved solubility and intestinal permeability, raising the possibility of using cisplatin as an oral drug. Moreover, cisplatin orally administered in rats after coating with nano-HAP showed equivalent anti-tumor effect to the intravenously administered drug, while reducing the level of renal and hepatic toxicity normally associated with cisplatin [18,19].

However, while demonstrating the ability of our nano-HAP coating to improve the solubility and intestinal permeability of drugs classified under BCS as poorly soluble and/or poorly absorbed, we had not yet elucidated the functional mechanism. Therefore, in the present study, we investigated the effect of nano-HAP coating on the BCS Class IV drug acetazolamide (AZ) [5,20], studying in particular its potential routes of intestinal absorption, using *in vitro*, *ex vivo* and *in vivo* methodology.

## Materials and methods

### Preparation of formulation and SEM observation

AZ was purchased from FUJIFILM Wako Pure Chemical Corporation (Osaka, Japan). The nano-HAP used for coating the surface of the AZ crystals in the AZ/HAP formulation was prepared as previously reported, including a final enteric coating needed for peroral use [19].

Briefly, chemically synthesized micron-level HAP (SKM-1, Sangi Co., Ltd) was ground with a wet grinding mill (DYNOMILL KDLA, Shinmaru Enterprises Co., Osaka, Japan) using zirconia beads 0.3mm in diameter and then dried. The primary particle size of the resulting powder was confirmed to be nano-scale by scanning electron microscope (SEM, Hitachi S-4500, Hitachi High-Tech Science Corporation, Tokyo, Japan), and the average particle size was approximately 130 nm as measured by laser scattering particle size distribution analyzer (LA-950; HORIBA, Ltd., Kyoto, Japan). Preparation of the AZ/HAP formulation was done by mechanical fusion (Mechanofusion AMS-Mini, Hosokawa Micron Group, Osaka, Japan) at 2,000 rpm for 15 min under an argon gas atmosphere. The weight ratio of AZ to nano-HAP in the AZ/HAP formulation was 1:2. The morphology of the AZ/HAP formulation was observed by SEM at an acceleration voltage of 15.0 kV. For animal testing, the AZ/HAP formulation was prepared with an enteric coating, using a mixture of acetone, cellulose acetate phthalate and diethyl phthalate at a weight ratio of 255:9:36.

## Solubility testing

Solubility testing was carried out using distilled water and disintegration test solution (DTS) 2 (phosphate buffer solution; pH 6.8) specified by the Japanese Pharmacopoeia. We did not use DTS 1 (phosphate buffer solution; pH 1.2) because the formulation in our study was an enteric-coated preparation. Fifty mL of each test solution was placed in a glass tube and stirred with a 15 mm teflon bar at a constant rotation speed of 200 rpm. Respectively, 300 mg of AZ and 900 mg of AZ/HAP formulation (containing 300mg of AZ) formulation were added to each of the two test solutions. One mL samples were collected into a microtube at 3, 10, 30, 120 and 360 min after the start of the test, which was performed in an incubator at $37 \pm 0.5°C$. Each sample was centrifuged for 3 min at $8,060 \times g$, and the supernatant freeze-dried. The dry sample was then dissolved in an appropriate amount of ethyl alcohol. The concentration of AZ in each sample was determined by spectrophotometer (UV-1200, Shimazu Corporation, Kyoto, Japan) at 264 nm wavelength. Solubility testing was carried out on AZ 3 times and on the AZ/HAP formulation 6 times.

## Pharmacokinetics study

Kwl:SD male rats (7 weeks; Tokyo Laboratory Animals Science Co., Ltd., Tokyo, Japan) were used in a pharmacokinetics study after single-dose oral administration of either the untreated AZ or AZ/HAP formulation. The dose of AZ in each case was 10 mg/kg, meaning that in the case of the AZ/HAP formulation 30 mg/kg of the physical mixture was administered. Rats were fasted for 16 h before oral administration. Each formulation was suspended in 5 mL water and administered by oral gavage using a rat feeding needle. Blood samples were collected in heparinized tubes from the tail vein at 0.5, 1.0, 3.0, 6.0 and 24.0 h after administration. Each sample was centrifuged for 15 min at $2,190 \times g$, and the supernatant stored at $-40°C$ until measured. The AZ group and the AZ/HAP group consisted of 4 and 7 rats, respectively. All animal experiments in the study were performed according to the guidelines for animal studies specified by the Science Council of Japan [21] and the internal regulations regarding animal experiments at Sangi Co., Ltd. Approval numbers provided by the Committee for Experimental Animal Care and Use at Sangi Co., Ltd. are as follows: 2018D3 (pharmacokinetics study), 2018D1(Ussing chamber study), and 2018D2 (localization in intestinal tissue using fluorescence-labelled HAP nanoparticles).

AZ in each sample was measured with reference to the papers of Ichikawa et al. [22] and Alberts et al. [23]. In brief, 100 μL of chlorothiazide (Tokyo Chemical Industry Co., Ltd., Tokyo, Japan) dissolved in water at a concentration of 50 μg/mL, 0.5 mL of 0.05 M sodium acetate (Special Grade, FUJIFILM Wako Pure Chemical Corporation) and 5 mL of ethyl acetate

(HPLC Grade, FUJIFILM Wako Pure Chemical Corporation) were added to each sample. Each tube was stirred with a vortex mixer for 5 min and centrifuged at $1,000 \times g$ for 10 min. The organic layer in the tube was transferred into a separate test tube and dried at room temperature under 100 mmHg atmosphere. 500 μL of the mobile phase solution used in HPLC was added to each dried sample. Finally, samples for quantitative analysis were prepared by passing them through a 0.45 μm filter (Minisart RC4; Sartorius AG, Göttingen, Germany).

The amount of AZ in plasma was determined by HPLC (PU-980, UV-970, UV/VIS detector, JASCO Corporation, Tokyo, Japan). YMC-Pack Pro C18 ($150 \times 4.6$ mm.I.D. S-5μm, 12nm, YMC Co., Ltd., Kyoto, Japan) was used as the separation column. The measurement was performed at room temperature using a flow rate of 1.0 mL/min, a measurement wavelength of 266 nm, and an injection amount of 100 μL. The mobile phase solution was a mixture of 0.05 M sodium acetate and acetonitrile (HPLC Grade, FUJIFILM Wako Pure Chemical Corporation) at 90:10 (V/V), and finally the pH was adjusted to 4.1 with acetic acid (Special Grade, FUJIFILM Wako Pure Chemical Corporation).The standard curve was calculated from the data of each concentration of AZ (0.5, 1.0, 2.5, 5.0, 10.0, 20.0 μg/mL).

Maximum concentration (Cmax) and AUC (0–6) values for both test groups after oral administration were calculated from the AZ blood concentration data obtained.

### Gel filtration chromatography analysis of AZ/HAP formulation

The components of the AZ/HAP formulation, namely AZ crystals and nano-HAP particles, each diffuse immediately upon contact with water. But it was unknown whether they exist independently in water, or are bonded. Therefore, we conducted an analysis using gel filtration chromatography.

PD-10 (Cytiva, Tokyo, Japan) filled with Sephadex G-25 (Medium) was used as the column for gel filtration chromatography, and the analysis was performed using DTS 2 and rat intestinal fluid. The intestinal fluid was collected from two 8-week-old male Kwl:SD rats (Tokyo Laboratory Animals Science Co., Ltd.) according to the method described by Alsulays et al. [24].

Two mL of the test solution was added to 10 mg of AZ, 20 mg of nano-HAP and 30 mg of the AZ/HAP formulation respectively, and each mixture was stirred at 37°C for 30 min. Each sample solution was then centrifuged at $8,060 \times g$ for 5 min, and 1.0 mL of the supernatant used for analysis. The PD-10 column was equilibrated with DTS 2 according to the manufacturer's instructions. One mL of each test sample was applied to the column, and elution performed by free fall. One mL of the eluate was sampled in the glass tube of a fraction collector (SF-160, ADVANTEC Co., Ltd., Tokyo, Japan), and the total elution volume was 20 mL.

500 μL of each sample was diluted 10-fold (in the case of DTS 2-treated samples) or 6-fold (in the case of intestinal fluid-treated samples) with ethanol solution (ethanol 3.5mL + water 1.0 mL) (ethanol, Special Grade, FUJIFILM Wako Pure Chemical Corporation), and the amount of AZ determined using a spectrophotometer (UV-1200, Shimadzu Corporation) at 364 nm wavelength. In addition, each sample was diluted 10-fold with 0.5 M nitric acid (Special Grade; FUJIFILM Wako Pure Chemical Corporation), and the amount of calcium determined using an inductively coupled plasma atomic emission spectrophotometer (SPS-1700R, Hitachi High-Tech Science Corporation). The amount of hydroxyapatite in each sample was calculated from the amount of calcium recorded.

### AZ and AZ/HAP formulation permeability test using a Caco-2 cell monolayer

A Caco-2 cell monolayer assay kit (POCA® Small Intestinal Absorption (Caco-2), KAC Co., Ltd., Kyoto, Japan) was used to test for the possible permeability of AZ and AZ/HAP

formulation via tight junctions, an application for which this apparatus is known [25]. The kit comprises an apical plate, perforated with holes that accommodate 24 wells, each well containing a Caco-2 cell monolayer at its base, to simulate the intestinal lumen, and a basolateral tray underneath, containing 24 receiving compartments into which the apical wells fit, to simulate the underlying tissues. The permeability test was carried out according to the instructions of the assay kit distributor, including the medium used. Transepithelial electrical resistance (TEER) was measured by epithelial voltmeter (EVOM2, World Precision Instruments Ltd., Hertfordshire, UK), and a $CO_2$ incubator (Direct Heat $CO_2$ Incubator model 320, Thermo Fisher Scientific, MA, USA) was used for culturing.

Test solutions were prepared by adding AZ or AZ/HAP formulation respectively to a Hank's balanced salt solution (HBSS) buffered with HEPES (4-(2-hydroxyethyl)-1-piperazineethanesulfonic acid, final concentration 25 mM) and $NaHCO_3$ (final concentration 0.35 g/L), at a concentration of 0.2 g AZ/mL. To ensure complete dissolution of the AZ crystals, dimethyl sulfoxide (DMSO) was added up to achieve a concentration of 0.5% [26]. Each test solution was stirred for 3 min and pre-warmed to 37°C before being added to the apical wells.

First, 1,000 μL of HBSS was added to each receiving compartment in the basolateral tray. The medium in each of the apical wells was then decanted off, and 200 μL of HBSS was added, after which the apical plate containing the wells was fitted into the basolateral tray, and equilibration carried out for 15 min at 37°C. Initial TEER was then assessed by measuring the electrical resistance (Ω) of the current flowing from the medium on the apical side to the medium on the basolateral side of the Caco-2 monolayers. The HBSS in each apical well was then decanted off and 200 μL of either test solution respectively was added. The kit was then incubated for culturing at 37°C for 1 hour, and TEER was again assessed, after which the entire amount of culture medium from each basolateral receiving compartment was collected.

The concentration of AZ in the respective samples collected was determined using HPLC according to the method described above. The apparent permeability coefficient (Papp) was calculated according to the following equation [27]:

$$Papp = (dQ/dt)(1/(AC_0))$$

where $dQ/dt$ is the steady-state flux (μg/s), $A$ is the diffusion area of each monolayer ($cm^2$) and $C_0$ is the initial concentration of the test drug (AZ) in each apical well (μg/mL). The percentage change in TEER value 60 min after addition of the respective test solutions, compared with the initial value, was also calculated.

The entire experiment was carried out three times, and the average of the results was calculated.

## Permeability testing using rat small intestine in an Ussing chamber

An *ex vivo* study using the Ussing chamber was carried out to investigate permeation via an intracellular route. An Ussing chamber is an apparatus for measuring epithelial membrane permeability for ions, nutrients, drugs etc., as shown in Fig 1. It consists of two baths separated by a sheet of mucosa or a monolayer of epithelial cells[28,29].

Since the properties of substances absorbed and the mechanisms of absorption differ between the region forming the intestinal villi and the lymphatic Peyer's patches in the intestinal wall [30–32], specimens of each tissue respectively were prepared for use in the Ussing chamber [28,29,33] and the permeability of the two drug solutions tested using each type.

The specimens were prepared from Kwl:SD male rats (8 weeks; Tokyo Laboratory Animals Science Co. Ltd.). Intestinal tissue was dissected from each rat after sacrifice by inhalation with isoflurane (MSD Animal Health K.K., Tokyo, Japan). Segments of intestinal mucosa for use in

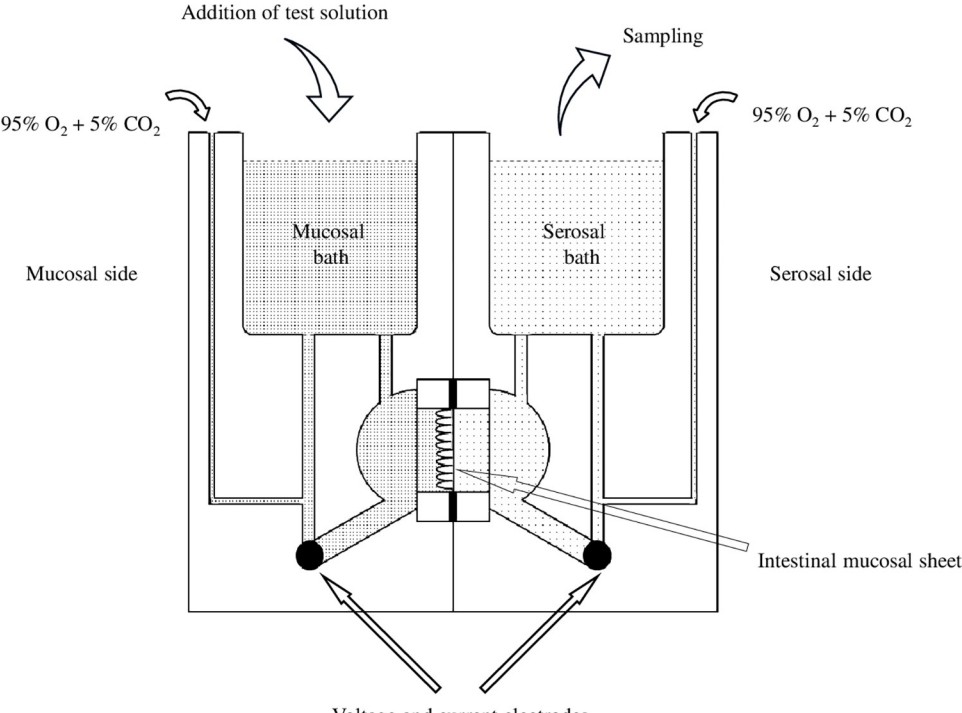

**Fig 1. Schematic diagram of an Ussing chamber (Authors' diagram).** The apparatus consists of two baths separated by, for example, a sheet of intestinal mucosa, one bath being on the mucosal side and the other on the serosal side. Each bath is filled with Krebs-Ringer solution kept at 37°C, and is constantly agitated by bubbling 95% $O_2$-5% $CO_2$ gas. The test drug solution is dropped into the mucosal bath, and any drug permeating into the serosal bath is measured upon collection after a given time.

the experiment (approximately 5 cm each) were taken from an area extending to about 50 cm below the duodenojejunal flexure. Tissue from the mucosal cell region not including Peyer's patches (NPP tissue) was prepared with reference to Clarke [28], keeping the epithelium and lamina propria intact, but removing the muscular layers. Tissue including a Peyer's patch (PP tissue) was prepared with reference to the method of Brayden and Baird [33]. All tissue preparation was designed to be completed within a 5-minute time-frame after euthanasia. Krebs-Ringer solution (117 mM NaCl, 4.7 mM KCl, 1.2 mM $MgCl_2$, 1.2 mM $NaH_2PO_4$, 25 mM $NaHCO_3$, 2.5 mM $CaCl_2$, 11 mM glucose, pH 7.3) was used for all experimental steps. Each tissue specimen was mounted in the Ussing chamber using a slider (P-2300 and P2305, window area = 0.49 $cm^2$ respectively, Physiologic Instruments, Inc., CA, USA). Both the mucosal and serosal baths were filled with 7 mL Krebs-Ringer solution kept at 37°C and stirred by continuous supply of 95% $O_2$-5% $CO_2$ gas. A total of 24 rats were used in the Ussing chamber experiments.

Ag-Ag/AgCl electrodes (P2020-S, Physiologic Instruments Inc.) were connected to the chamber, and TEER was measured by epithelial voltmeter. TEER values were used as evaluation criteria for both NPP and PP tissue to determine whether each excised specimen was functioning normally [34–36]. If the TEER value was below 70 $\Omega cm^2$ in NPP tissue and below 65 $\Omega cm^2$ in PP tissue, the tissue was not used in the experiment. In the PP tissue, whether or not the tissue was functioning was also evaluated by a macromolecular permeability test using horseradish peroxidase (HRP type VI; Sigma-Aldrich Co. LLC, STL, USA) with reference to

Keita et al. [37]. HRP was determined using the Pierce™ TMB ELISA Substrate Kit (Thermo Fisher Scientific).

Each specimen, after mounting in the Ussing chamber, was left idle for 15 min. Since the ratio of AZ to HAP was 1: 2 w/w for the AZ/HAP formulation, 1/3 of the HAP formulation was calculated as AZ. The AZ and AZ/HAP test solutions were prepared as follows.

Whether using untreated AZ or AZ/HAP formulation, the amount added to 500 μL of Krebs- Ringer solution for use in the mucosal bath was adjusted to a final concentration of either 540 ppm or 2,240 ppm of AZ, and the suspension was mixed for 1 min by vortex mixer. 500 μL of each test solution respectively was then added to the mucosal bath after removing 500 μL of Krebs-Ringer solution from the bath in order to equalize the water pressure between the two baths. Similarly, 500 μL of solution was taken from the serosal bath and the same volume of fresh Krebs-Ringer solution added at 30, 60, 90, and 120 min, and four samples were collected at each time point. Each sample solution was immediately transferred to a plastic tube and stored at −80˚C until the AZ was measured. Each frozen solution was thawed at room temperature, passed through a 0.45 μm filter, and 10 μL used for HPLC measurement of AZ under the conditions previously shown.

## Intestinal localization of nano-HAP fluorescence-labelled particles

HAP nanoparticles are known to be promising carriers for drugs and nucleic acids in drug delivery systems [38,39], but their routes of entry into the tissues remain unclear. Since information on the localization of the nano-HAP particles in the intestinal tract after oral administration could provide useful clues for identifying the absorption pathway of AZ in the AZ/HAP formulation, the location of particles labeled with two types of fluorescent dye was investigated.

**Fluorescence labeling of nano-HAP particles.** Two types of fluorescent dye were used to label nano-HAP particles for observation in the intestinal tissue. One was Calcein (Research Grade; FUJIFILM Wako Pure Chemical Corporation), a calciphilic fluorescent dye [40], and the other OsteoSense 680EX (PerkinElmer Inc., MD, USA), a hydroxyapatite-affinity fluorescent dye used by the authors in a previous paper [41].

First, nano-HAP and Calcein were mixed, at a ratio of 1:25 w/w, with an appropriate amount of distilled water and stirred at room temperature for 2 hours. The HAP-Calcein complex was then washed by a successive process of centrifugation-decantation-washing to remove Calcein not bound to nano-HAP particles. The mixture was transferred to a plastic tube and centrifuged at 8,060 × g for 10 min. After removal of the supernatant, an appropriate amount of distilled water was added to the tube containing the precipitate, and the mixture was treated with an ultrasonic generator (USC-100Z38S-22; Ultrasonic Engineering Co., Ltd., Tokyo, Japan) in 120 W for 5 min. Fluorescence was observed in the supernatant using an epifluorescence microscope with a mercury lamp (BX60 and U-ULH respectively; Olympus Corporation., Tokyo, Japan). The further washing process was repeated until no fluorescent dye was observed in the supernatant and the precipitate was then suspended in an appropriate amount of distilled water. Subsequently, OsteoSense 680EX was added to this suspension at a ratio of 0.08 nmol to nano-HAP 30 mg, and the mixture stirred at room temperature for 3 hours. The centrifuge-decant-wash process was repeated three times to remove OsteoSense 680EX not bound to nano-HAP particles.

**Localization of fluorescence-labeled nano-HAP particles in the rat small intestine.** Kwl:SD male rats (8 weeks; Tokyo Laboratory Animals Science Co., Ltd.) were used for localization of nano-HAP particles labeled with Calcein and OsteoSense 680EX. Rats were fasted and given water ad libitum 16 hours before oral administration of fluorescence-dyed nano-

HAP particles. The nano-HAP particles were suspended in water and orally administered at 300 mg/10 mL/kg. Only water was administered to the untreated controls. Each group consisted of 3 rats.

Rats were euthanized by over-anesthesia with isoflurane 30 min after oral administration. After confirming death, the region from the jejunum to the upper ileum, the small intestine including Peyer's patches and the liver were removed and cut to an appropriate size. Each tissue sample was fixed by immersing in a 4% paraformaldehyde/PBS(−) solution statically at 4˚C overnight and then immersed in PBS(−) solution statically at 4˚C for 1 hour. Each sample was cut into appropriate blocks and then embedded in Tissue Tech® O.C.T. compound (Sakura Finetek Japan Co., Ltd., Tokyo, Japan) using liquid nitrogen. Each block was sliced to a thickness of 5 μm at −20˚C using a cryostat (Leica CM3050S; Leica Microsystems Inc., Wetzlar, Germany) and attached to a glass slide (Platinum Pro; Matsunami Glass Ind., Ltd., Osaka, Japan). Each frozen section was enclosed with SlowFade™ Diamond Antifade Mountant using DAPI fluorescent dye for nuclear staining (Thermo Fisher Scientific K.K., Tokyo, Japan).

Each section was observed using a confocal laser scanning microscope (TCS-SP5 II; Leica Microsystems Inc.) at a magnification of × 630. The excitation wavelengths of OsteoSense 680EX, Calcein and DAPI were 633, 490 and 405 nm respectively, and the fluorescence wavelengths were $690 \pm 10$, $510 \pm 15$ and $460 \pm 15$ nm respectively.

## Statistical analysis

Data were expressed as the mean ± S.D. Means were compared by Student's t-test to identify differences between groups. A value of $p < 0.05$ was accepted as significant.

## Results

### SEM observation

Fig 2 shows SEM micrographs of AZ (Fig 2A) and the AZ/HAP formulation (Fig 2B). The crystal surface of AZ was smooth, and the crystal size measured from micrographs was approximately 20 μm. In contrast, the AZ/HAP formulation particles were slightly smaller, and comprised AZ almost entirely covered by HAP nanoparticles.

### Solubility testing

Fig 3 shows the results of solubility testing of AZ and the AZ/HAP formulation, (Fig 3A) in water and (Fig 3B) in DTS 2. In the test with water, the amount of AZ dissolved increased

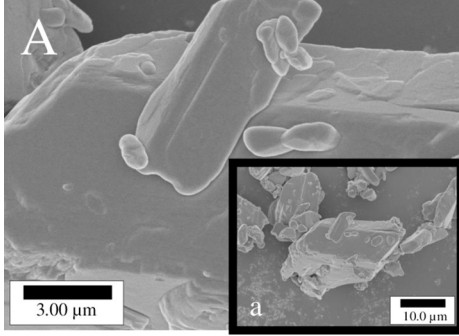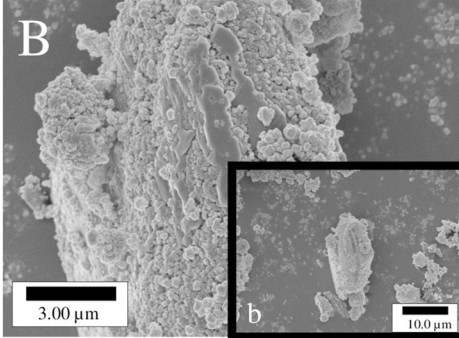

**Fig 2.** SEM micrograph of AZ crystals (A) and AZ/HAP formulation (B). A and B are enlargements of photographs a and b, respectively.

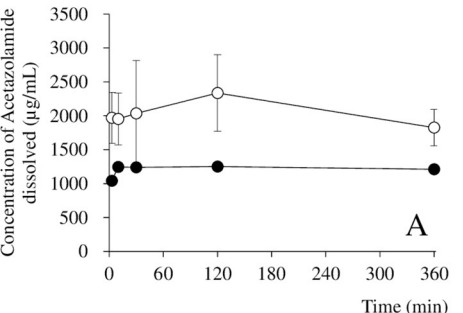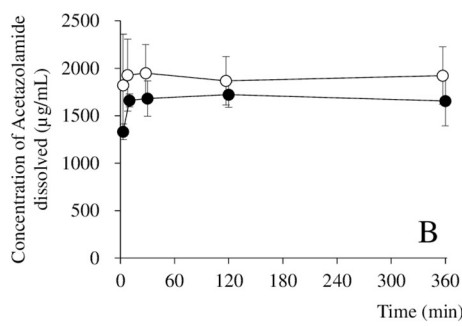

**Fig 3.** Solubility testing of AZ and AZ/HAP formulation in water (A) and DTS 2 (B). Samples were taken at 3, 10, 30, 120, and 360 min from the start of the test. The black circles and white circles indicate AZ and AZ/HAP formulation, respectively. Each value is the mean ± S.D. obtained after testing AZ 3 times and the AZ/HAP formulation 6 times.

slightly at 3 min, then plateaued, and no further change was observed for up to 360 min. Error bars are not visible in the case of AZ alone as the standard deviation was extremely small. In the case of the AZ/HAP formulation, the amount of AZ dissolved at 3 min was about twice that for AZ alone, and it gradually increased up to 120 min, then continued to decline thereafter up to 360 min. In the test with DTS 2, the amount of AZ dissolved in the case of both test solutions increased significantly for up to 10 min, but showed no further increase thereafter for up to 360 min. No significant difference in AZ solubility between the two groups at any sampling time was observed.

## Pharmacokinetics study

The blood concentrations of AZ measured for the AZ group and the AZ/HAP formulation group at each sampling time are shown in Fig 4. Blood concentration in the AZ group increased relatively slowly, peaking at 1 hour after administration, in contrast with the AZ/HAP formulation group, in which blood concentration was significantly higher, increasing rapidly within the first 30 min after administration then continuing to decline slowly thereafter. It decreased to a level that was not significantly different from that of the AZ group at 6 hours.

Table 1 shows Cmax and AUC (0–6) calculated from the blood concentration values of AZ in each drug administration group. The Cmax of the AZ/HAP formulation group was about 5 times higher than that of the AZ group. The AUC (0–6) value was about 4 times higher in the AZ/HAP formulation group than in the AZ group.

## Gel filtration chromatography analysis of AZ/HAP formulation

The results of gel filtration chromatography carried out on nano-HAP and untreated AZ, and on AZ/HAP formulation treated with either DTS 2 or with intestinal fluid are shown in Fig 5. The recovery rate was almost 100% in all elution experiments. In the case of the untreated AZ solution (Fig 5A), AZ was first observed from 4 mL, and the maximum amount was observed at 7 mL, while in the case of the nano-HAP suspension (Fig 5B), nano-HAP was observed from 1 mL, and already showed its maximum elution rate at 2 mL. In the case of the AZ/HAP formulation treated with DTS 2, the elution pattern of AZ (Fig 5C) was almost the same as that of AZ without DTS 2 treatment, and the elution pattern of nano-HAP (Fig 5D) was also almost identical with that of nano-HAP without DTS 2 treatment (Fig 5B). In the case of the AZ/HAP formulation treated with intestinal fluid, a small peak was observed in which AZ was eluted slightly earlier than in Fig 5A and 5C (Fig 5E), while for nano-HAP, the initial peak appeared to be divided into two, showing a broad overall elution pattern (Fig 5F).

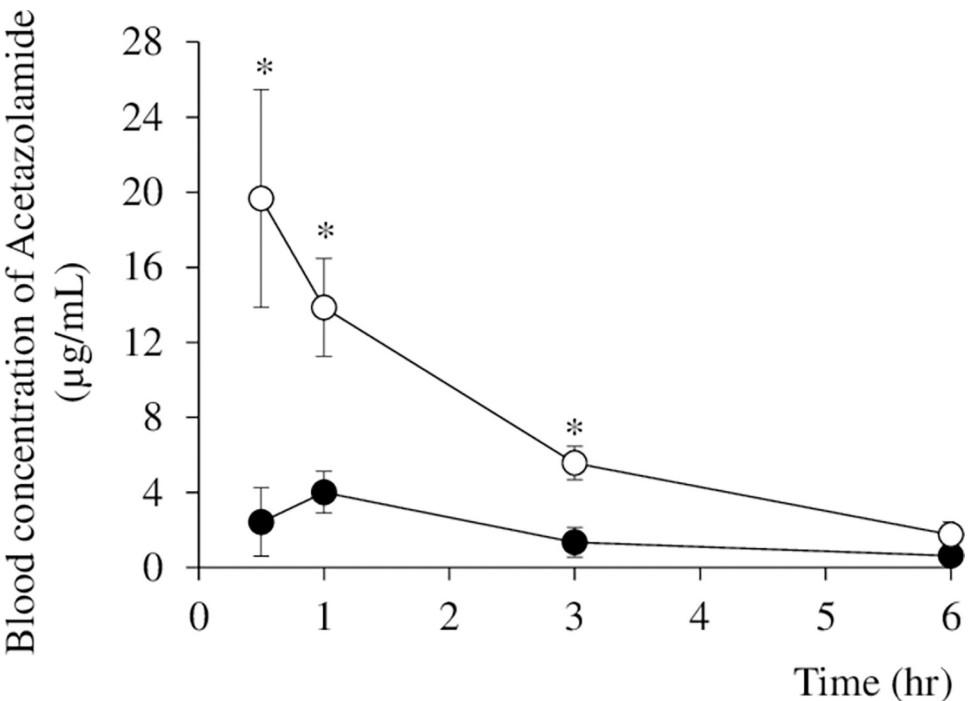

**Fig 4. Levels of blood concentration of AZ after oral administration of untreated AZ and AZ/HAP formulation in rats.** Samples were taken at 0.5, 1.0, 3.0 and 6.0 hours from the start of the test. Four rats were used in the AZ group (black circles) and seven in the AZ/HAP formulation group (white circles). Each value is the mean ± S.D. obtained. Asterisks show a significant difference (p <0.05) between the AZ group and AZ/HAP group using Student's t-test.

## AZ and AZ/HAP formulation permeability test using a Caco-2 cell monolayer

Fig 6 shows the result of permeability testing of AZ and the AZ/HAP formulation using a Caco-2 cell monolayer assay kit. The vertical axis shows the Papp value. Although the average value of Papp was somewhat higher in the AZ/HAP formulation group no statistical difference between the two groups was observed. Table 2 shows the percentage change in TEER value 60 min after addition of each respective test solution, compared with the initial value. This level of change is not considered to indicate damage to cells resulting from any external cause or substance [42,43], and it was not significantly different between the two test groups.

## Permeability testing using rat small intestine in an Ussing chamber

**Permeability in NPP tissue.** Fig 7 shows the results of testing for permeability of AZ across NPP tissue when respectively using AZ and the AZ/HAP formulation. When the final

**Table 1. Cmax and AUC (0–6) values obtained from the blood concentration of AZ in each drug administration group.**

| Group | Cmax (µg/mL) | AUC (0–6) (µg/mL·h) |
|---|---|---|
| AZ (n = 4) | 4.00 ± 0.60 | 10.52 ± 1.87 |
| AZ/HAP formulation (n = 7) | 19.65 ± 5.81* | 43.70 ± 8.24* |

Each value is the mean ± S.D. obtained. Asterisks show a significant difference (p <0.05) between the AZ group and AZ/HAP group using Student's t-test.

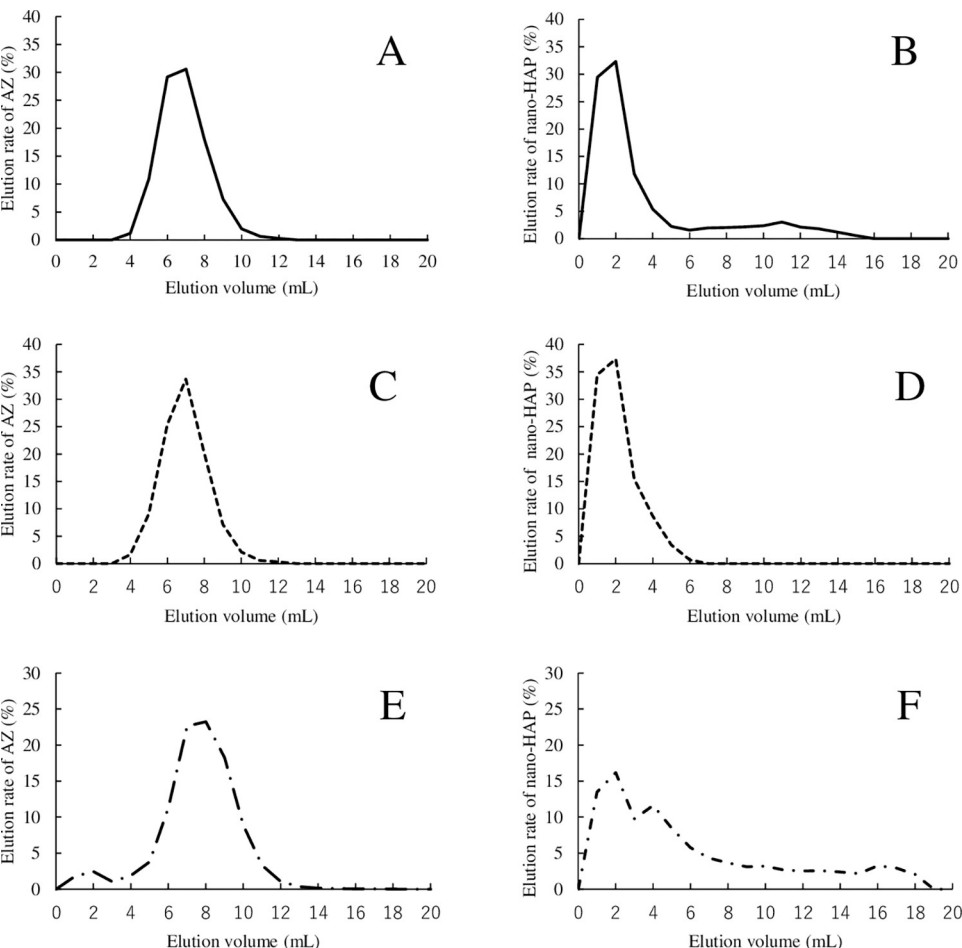

**Fig 5. Elution patterns of gel filtration chromatography for AZ, nano-HAP and AZ/HAP formulation.** Graphs A and B show the elution patterns of AZ and nano-Hap obtained from AZ solution and from nano-HAP suspension treated with DTS 2 on the gel filtration column, respectively. C and D show the respective elution patterns of AZ and nano-HAP obtained from the analysis of AZ/HAP formulation treated with DTS 2. E and F show the respective elution patterns of AZ and nano-HAP obtained from the analysis of AZ/HAP formulation treated with intestinal fluid. The horizontal axis shows the amount of elution, and the vertical axis shows the elution rate (%) of each component.

concentration of AZ in the mucosal bath after addition of each test solution was 560 ppm (Fig 7A), the degree of AZ permeation was shown to increase over time in both groups, but was small and not significantly different between the groups. When the final concentration of AZ in the mucosal bath was raised to 2,240 ppm (Fig 7B), the degree of AZ permeation was about 5 times higher in both groups, and it also increased over time, but level of permeation was significantly higher at all sample times for the AZ/HAP formulation group. Values for TEER did not rise during the experiment and no significant difference in them was observed between the two groups, regardless of the initial concentration of AZ.

**Permeability in PP tissue.** Fig 8 shows the results of testing for permeability of AZ across PP tissue when respectively using AZ and the AZ/HAP formulation. In the case of PP tissue, whether the final concentration of AZ in the mucosal bath was 560 ppm (Fig 8A) or 2,240 ppm (Fig 8B), the degree of AZ permeation observed for both groups once again increased over time and was about 5 times higher at the higher concentration level, but there was no statistically significant difference in permeation between the AZ group and the AZ/HAP formulation

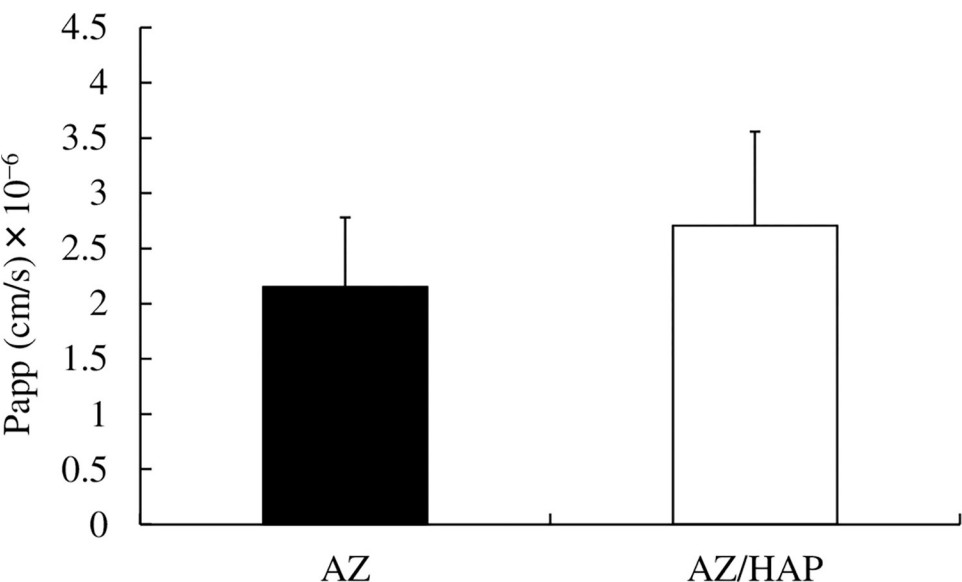

**Fig 6. Permeability assay of AZ and the AZ/HAP formulation using a Caco-2 cell monolayer kit.** The vertical axis indicates Papp value (cm/s). The bar above each column indicates S.D. No significant difference between the AZ group and the AZ/HAP formulation group was observed.

group. Values for TEER did not rise during the experiment, and no significant difference in TEER between the two groups was observed.

In macromolecular permeability testing prior to the trial, using horseradish peroxidase to determine whether the PP tissue mounted in the Ussing chamber retained the structure and function of follicle-associated epithelium, our results showed that the extracted tissue was functioning normally. We therefore considered our technique sufficient to produce specimens that could be viably used in this study.

### Intestinal localization of nano-HAP-fluorescent particles

Nano-HAP particles labeled with two types of fluorescent dyes were used to investigate the location of intestinal absorption after oral administration of the particles. Tissue photographs taken 30 min after administration are shown in Fig 9. No fluorescence was observed at the fluorescence wavelengths in control rats treated only with water, indicating that the fluorescence of OsteoSense and Calcein observed in this experiment was not affected by any factor of autofluorescence. In addition, as shown in smaller photographs in the right column, the respective fluorescences of OsteoSense 680EX and Calcein were located in the same position, therefore we concluded that these fluorescences were from nano-HAP particles.

Fluorescence observation showed that nano-HAP particles were already located in the epithelial cells of the villi and in the submucosal tissue 30 min after oral administration (Fig 9A).

**Table 2. Change in TEER Value (%).**

| Group | TEER (% of initial value) |
|---|---|
| AZ | 89.2 ± 1.9 |
| AZ/HAP formulation | 91.8 ± 2.2 |

Each value is the mean ± S.D. obtained. Figures show % change in TEER value 60 min after addition of each test solution relative to the initial value. This level of change is not considered to indicate damage to cells.

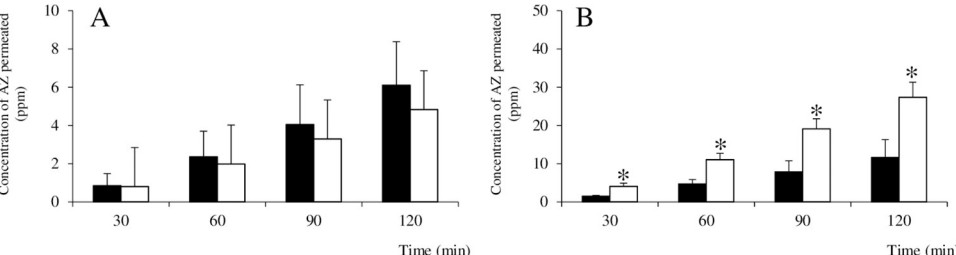

**Fig 7. Permeability of AZ through NPP tissue using an Ussing chamber.** Graph A shows the amount of AZ at each sampling time in the serosal bath when the final concentration of AZ in the mucosal bath was 560 ppm. Graph B shows the results when the final concentration of AZ in the mucosal bath was 2,240 ppm. Black and white columns show results for the AZ and AZ/HAP formulation groups respectively. Each value is the mean ± S.D. obtained (n = 4). Asterisks indicate a statistically significant difference (p <0.05) between the two groups. Note that A and B have different vertical scales.

The presence of many nano-HAP particles was observed in Peyer's patches (Fig 9B), and scattered fluorescence was also seen within the sinusoids of the liver (S1 Fig).

## Discussion

We previously reported that for some poorly water-soluble drugs, coating with nano-HAP particles improved bioavailability and reduced toxicity [15,18,19]. In the present study, we succeeded in improving the bioavailability of acetazolamide (AZ), a BCS Class IV low-solubility, low-permeability drug, by mechanically coating its crystal surface with nano-HAP particles.

The target of this HAP formulation is poorly water-soluble drugs, namely BCS Class II and Class IV drugs. In the case of BCS Class II drugs, which often show acidity, adjusting the pH of the drug in solution to near neutral is known to increase the solubility of the drug and therefore its absorption from the intestinal tract [10,11]. HAP is known to function as a buffer in an acidic pH range [44]. In an earlier study, we succeeded in increasing the AUC of the BCS Class II drug bezafibrate by using a HAP formulation [15], and we postulated that this buffer function of HAP may play a part in improving the drug's solubility. In the present study, however, although the BCS Class IV drug AZ was uniformly coated with nano-HAP particles, its solubility in DTS 2 was not significantly improved. AZ in solution is reported to be only weakly acidic, with a pKa of 7.2 [20], and this weak acidity might conceivably have precluded any positive buffering effect of HAP.

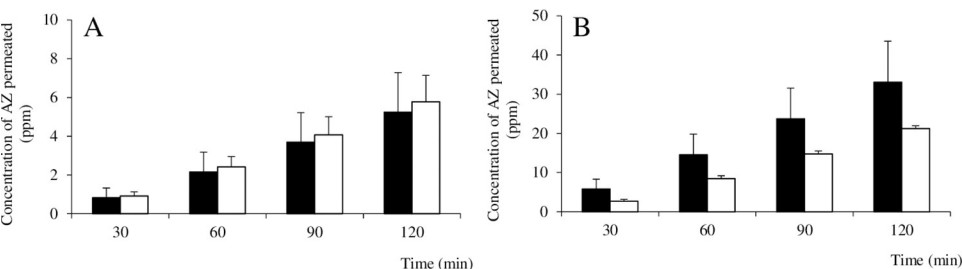

**Fig 8. Permeability of AZ through PP tissue using an Ussing chamber.** Graph A shows the amount of AZ in the serosal bath at each sampling time when the initial concentration of AZ in the mucosal bath was 560 ppm. Graph B shows the results when the final concentration of AZ in the mucosal bath was 2,240 ppm. Black and white columns show results for the AZ and AZ/HAP formulation groups respectively. Each value is the mean ± S.D. obtained (n = 4). Note that A and B have different vertical scales.

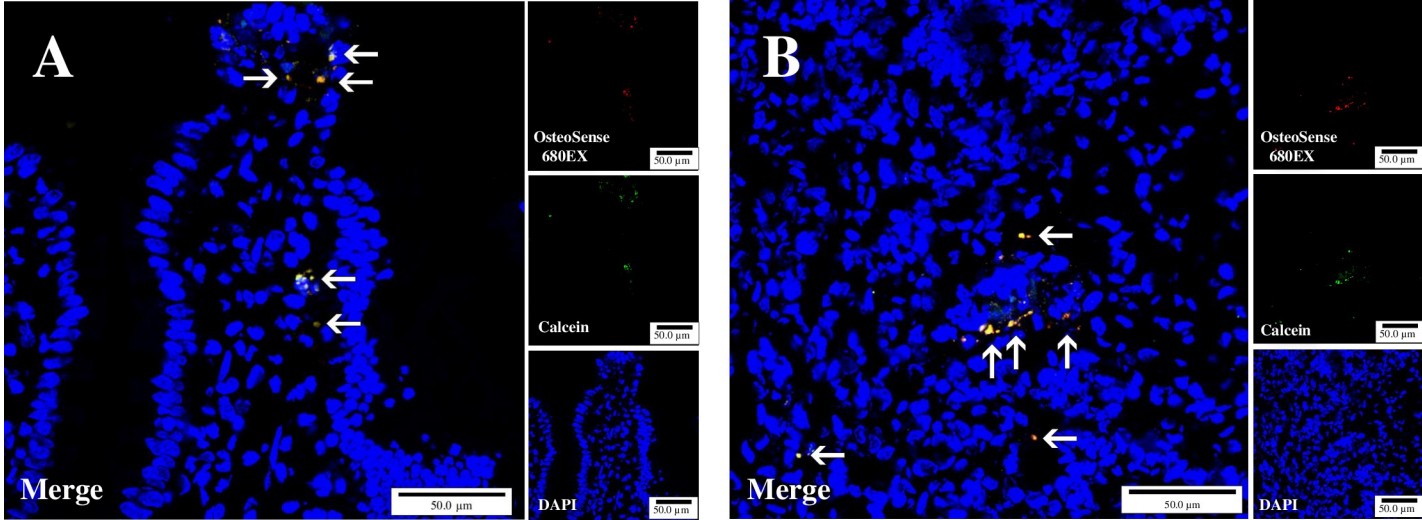

**Fig 9.** Localization of OsteoSense 680EX and Calcein double-labeled nano-HAP particles in rat intestinal villi (A) and Peyer's patches (B) 30 min after oral administration. Photos in the right column show the fluorescence observed at each wavelength for OsteoSense 680EX (top), Calcein (middle), and DAPI (bottom), respectively. The photo on the left (Merge) is a layered version of these three photos. Arrows indicate the location of the nano-HAP particles seen with fluorescence.

Since AZ is a BCS Class IV drug, an improvement in bioavailability could not have been expected from the results of our solubility testing, which showed that both AZ and the AZ/HAP formulationwere poorly soluble. Yet after single-dose oral administration, the AZ/HAP formulation group showed a Cmax about 5 times higher and an AUC (0–6) about 4 times higher than for the AZ group, strongly suggesting that the nano-HAP particles play a role in the absorption mechanism.

Although various methods for improving the bioavailability of BCS Class IV drugs have been reported [10,11], the use of mechanical coating of nano-HAP particles on the drug surface has not been among them. Nano-HAP particles have been widely studied as potential drug carriers, especially for their benefits of slow release and lesion targeting, and are expected before long to be put into practical use [39,40,45,46]. We initially expected that the nano-HAP particles in the HAP formulation would remain combined with AZ in the intestinal tract, and that this nano-HAP-AZ complex would be absorbed by a route different from that of AZ, with the result that the blood concentration of AZ would increase. However, gel filtration analysis of the AZ/HAP formulation, whether treated with DTS 2 or intestinal fluid, showed that AZ was not bound to the nano-HAP particles.

When treated with intestinal fluid, the nano-HAP particles showed biphasic separation peaks, unlike those seen when the particles were tested with DTS 2, and the overall elution pattern moved posteriorly. In gel filtration chromatography analysis, larger molecules elute first and smaller molecules later [47]. Results in the present study showed that the nano-HAP particles aggregated in the DTS 2, but tended to disperse in the intestinal fluid. A protein corona is reported to form on polystyrene nanoparticles of 50–100 nm when they come into contact with digestive enzymes such as α-amylase and trypsin [48], and proteins contained in intestinal fluid are also known to cause aggregation of nanoparticles [49]. Our results provided no direct information regarding protein corona formation or particle aggregation. However, it is possible that any aggregates of nano-HAP particles that did form may have tended to disperse under the influence of bile in the intestinal fluid, which is known to act as a dispersant [50], and that nano-HAP in the AZ/HAP formulation may affect the absorption mechanism of AZ in the form of dispersed particles rather than as aggregates.

The main absorption pathways for nutrients and drugs in the intestinal tract are the intercellular and intracellular pathways of the epithelium, while many large molecules and/or insoluble substances are absorbed via the lymphatic system [51]. The tight junction forms a barrier between cells, blocking the paracellular pathway, through which many substances are nevertheless absorbed by passive transport.

The intracellular pathway involves transport using various channels and absorption systems that are intricately intertwined [52–54]. These include many types of membrane transporters, as well as ATP-binding cassette transporters, P-glycoprotein and other entities involved in the absorption and excretion of drugs and other substances [52–54]. Membrane transport may be due to morphological changes in cell membranes, including systems such as endocytosis and exocytosis, or phagocytosis and pinocytosis [5,54], or it may result from damage to the mucosal epithelium caused by certain drugs or pathologies [55]. There is also a mechanism known as direct permeation in which substances move directly into the cell as a result of a particular stimulus [56–58].

While most substances entering the body from the small intestine are taken up via the intestinal villi, as an alternative route, especially for the entry of large molecules, substances not absorbed via the villi may enter via the Peyer's patch tissues [51,59]. These contain cells called microfold cells (M cells) that are known to absorb large substances by endocytosis on the mucosal side of the cell and expel them from the basal membrane side in a transcellular process described as transcytosis [60]. It is also known that calcium phosphate nanoparticles formed endogenously in the small intestine can trap soluble macromolecules such as protein antigens in the lumen and transport them for removal via the M cells in the follicle-associated membrane covering the Peyer's patches, through which the macromolecule-containing nanoparticles are absorbed, initiating an antigen-specific mucosal immune response [61].

In order to explore and elucidate the mechanism of increased absorption of AZ seen with the AZ/HAP formulation, experiments involving a large number of detailed pathways are required. As a first step in these studies, we examined which of three pathways—intercellular, intracellular, and lymphatic—may be involved.

To begin with, we looked at the intercellular pathway, in which tight junctions play a barrier role. For this research we selected the Caco-2 cell monolayer assay kit, which has long been used for tight junction research [25], focusing especially on the electrical resistance between cells (TEER), changes in which can be a sign of cell damage and/or of junction opening. As an electrical phenomenon, this resistance can be measured when electrodes are placed on both sides of a cell sheet [34]. Tight junction opening, for example, can be determined by a large, sharp drop in TEER value [42,43]. TEER values are also known to correlate highly with cell viability, so cell damage can be assumed if there is a greater fluctuation in TEER value than when tight junctions are affected [35,36].

Testing in the present study using the Caco-2 cell monolayer assay kit showed no significant difference in permeability between the AZ/HAP formulation group and the AZ group. There was almost no change in TEER value in either case and therefore no evidence of either opening of the tight junctions or of cell damage during the experiment. We therefore postulated that the nano-HAP particles in the HAP formulation do not enter the tissues by causing opening of tight junctions, as chitosan reportedly does [62,63], or by damaging cells as titanium oxide nanoparticles are reported to do [64]. Our results suggested that nano-HAP particles enter the tissues via intracellular pathways rather than the intercellular route. Moreover, although in recent years, concern has arisen over possible nanoparticle cytotoxicity [65], in the case of nano-HAP in the present study, no evidence of cytotoxicity was observed.

We therefore next conducted an experiment using the Ussing chamber as a test for intracellular transport. In contrast to the *in vitro* single cell-culture Caco-2 model, the *ex vivo* Ussing

method uses fresh intestinal epithelium consisting of various functional cells, offering more complex physiological interaction [66]. Another advantage of this method is that in addition to a sample of intestinal epithelium containing principally villi, a sample containing principally Peyer's patch tissue can also be used [33,67]. In the case of poorly water-soluble drugs, there is some debate about the possibility that drug solids not completely dissolved in the intestinal tract after oral ingestion may be absorbed as they are [5,68,69], with transport by intracellular pathways or the lymphatic system considered to be the most probable routes.

In tests with the Ussing chamber, using intestinal epithelium both with and without Peyer's patch tissue, both low and high concentrations of AZ or AZ/HAP formulation were suspended in Krebs-Ringer solution and then dropped into the mucosal bath, and as DMSO was not used, this may have caused crystals that were not yet completely dissolved to come into contact with the intestinal epithelium. In both types epithelium, at low concentration, no significant difference in AZ permeability between the test groups was observed, but at a high concentration of AZ, the AZ/HAP formulation group showed significantly greater AZ permeability than the AZ group. Since the pH of the Krebs-Ringer solution in the Ussing chamber is neutral, and our prior testing showed almost no difference in solubility betweenAZ and the AZ/HAP formulation, the reason for this difference in permeability remains unclear. However, as in the Caco-2 cell monolayer test, no significant changes in TEER were observed for either group at any time point, indicating that no cellular damage or destruction by AZ [70] or the nano-HAP particles had occurred.

It is known that HAP is highly biocompatible and has a strong propensity for cellular adhesion [71]. Therefore, it is possible that adhesion of nano-HAP to the cell surface could interfere with the systems involved in mass transport, in which case the amount of drug absorbed could be expected to decrease. Conversely, inhibition of the ATP-binding cassette transporters and their family of P-glycoproteins, which are known to be involved in drug excretion [52], could result in increased absorption of their substrate drug. It has been shown that AZ is a substrate for P-glycoprotein and that the rate of AZ efflux from Caco-2 monolayers was three times higher than the uptake rate, but that P-glycoprotein inhibitors significantly reduced this rate of efflux [20]. Because of its high affinity for protein, HAP is often used for chromatographic purification of proteins, not only secreted from cells but also expressed on their surface [72], and ceramic HAP chromatography has been successfully used for the purification of P-glyco-protein [73], indicating that HAP has the potential to bind to this protein. In our experiment using NPP tissue in the Ussing chamber, at a high level of initial AZ concentration, AZ permeability was significantly higher for the AZ/HAP formulation than for AZ alone, while at a lower level of initial concentration, no significant difference in AZ permeation between the two groups was observed. This raises the interesting possibility that nano-HAP may have an inhibitory effect on P-glycoprotein functioning, and might possibly contribute to AZ absorbance by interfering with drug efflux and/or protein synthesis in intestinal epithelial cells. More work to confirm this hypothesis however will be necessary.

In cell membrane transport, endocytosis is primarily involved in the transport of solid particles, and nano-HAP particles are known to be taken up into cells by endocytosis [74]. Indomethacin nanoparticles have also been reported to be incorporated into the intestinal epithelium by energy-dependent endocytosis [65]. If an increase in endocytosis, due to the presence of nano-HAP particles, also results in increased endocytosis of other substances, it is possible that any AZ crystals not yet dissolved might be absorbed into cells.

In any case, it was not possible to clearly show the cause of the increased absorption of AZ from the intestinal tract observed in the AZ/HAP formulation group in this study. It is possible that there may be more than one mechanism operating to increase AZ permeability. However, our results suggest strongly that nano-HAP particles affect not tight junctions but rather

aspects of the intracellular pathway. Continuing studies to clarify the mechanism of absorption of drugs mechanically coated with nano-HAP particles may lead to the discovery of new biological functions of HAP.

## Conclusion

BCS Class IV drugs face the problem of low bioavailability because of their low solubility and low permeability. We succeeded in improving the bioavailability of acetazolamide (AZ), a BCS Class IV drug, when orally administered, by mechanically coating its surface with nano-HAP particles, even though there was no difference in solubility between AZ and the AZ/HAP formulation.

In the AZ/HAP formulation, AZ and nano-HAP particles were shown by gel filtration chromatography analysis not to be bound, either in phosphate buffer solution (DTS 2) or intestinal fluid. Therefore, it was considered that nano-HAP particles did not act as carriers of AZ but were nevertheless effective in increasing the drug's absorption from the intestinal tract.

In the present study, testing with a Caco-2 cell monolayer assay kit indicated that nano-HAP was not causing any opening of tight junctions or any damage to cells. Experiments using the Ussing chamber strongly suggested that nanoparticles' effect on drug permeability instead involves intracellular pathways of the intestinal epithelium, and to some extent the lymphatic system, via the Payer's patches. At a high level of drug concentration in testing with intestinal epithelium not containing Peyer's patch tissue, drug permeability was significantly higher in the case of the AZ/HAP formulation that for AZ alone.

The method of mechanically coating a drug with nano-HAP particles presented in this paper is simple and does not cause any chemical reaction or alteration to the drug in the manufacturing process. We believe it to be a unique way of improving the bioavailability of poorly water-soluble, poorly absorbable drugs.

In the present study, however, we could not elucidate the mechanism of nano-HAP's effect on the absorption of the BCS Class IV drug from the intestinal tract, and more detailed research on this topic will be required.

## Supporting information

**S1 Fig. Localization of OsteoSense 680EX and Calcein double-labeled nano-HAP particles in rat liver 30 min after oral administration.** (A) Photos show the fluorescence observed at each wavelength for DAPI, Calcein, OsteoSense 680EX, and Merge, respectively. Arrows indicate the location of the nano-HAP particles seen with fluorescence. The region surrounded by dashed lines in the merge image has been enlarged in S1B Fig. (B) Enlarged images of S1A Fig. (TIF)

**S1 File. Minimal data set.** Data underlying our results in this manuscript. (XLSX)

## Acknowledgments

We thank Tokyo Metropolitan Industrial Technology Research Institute for their excellent technical support in laser microscopic observation, and former colleague Dr. Keiichiro Kikukawa for his expert advice concerning the manuscript.

## Author Contributions

**Conceptualization:** Roslyn Hayman.

**Data curation:** Kenichi Kaneko, Ryosuke Miyasaka.

**Formal analysis:** Kenichi Kaneko, Ryosuke Miyasaka.

**Investigation:** Kenichi Kaneko, Ryosuke Miyasaka.

**Methodology:** Kenichi Kaneko, Ryosuke Miyasaka.

**Project administration:** Ryosuke Miyasaka.

**Supervision:** Roslyn Hayman.

**Validation:** Kenichi Kaneko, Ryosuke Miyasaka.

**Visualization:** Kenichi Kaneko, Ryosuke Miyasaka.

**Writing – original draft:** Kenichi Kaneko.

**Writing – review & editing:** Ryosuke Miyasaka.

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
