## [Decision Letter · Decision Letter 0]

2 Feb 2022

PONE-D-21-40400Nano-hydroxyapatite improves intestinal absorption of acetazolamide (BCS Class IV drug) – but how?PLOS ONE

Dear Dr. Kaneko,

Thank you for submitting your manuscript to PLOS ONE. After careful consideration, we feel that it has merit but does not fully meet PLOS ONE’s publication criteria as it currently stands. Therefore, we invite you to submit a revised version of the manuscript that addresses the points raised during the review process.

We look forward to receiving your revised manuscript.

Kind regards,

Vivek Gupta

Academic Editor

PLOS ONE

Journal Requirements:

"This study was funded by Sangi Co., Ltd. The funder provided support in the form of salaries for authors (Kenichi Kaneko and Ryosuke Miyasaka) as well as research funds. The funder had no role in the study design, data collection and analysis, decision to publish, or preparation of the manuscript."

We note that one or more of the authors is affiliated with the funding organization, indicating the funder may have had some role in the design, data collection, analysis or preparation of your manuscript for publication; in other words, the funder played an indirect role through the participation of the co-authors. If the funding organization did not play a role in the study design, data collection and analysis, decision to publish, or preparation of the manuscript and only provided financial support in the form of authors' salaries and/or research materials, please do the following:

a. Review your statements relating to the author contributions, and ensure you have specifically and accurately indicated the role(s) that these authors had in your study. These amendments should be made in the online form.

b. Confirm in your cover letter that you agree with the following statement, and we will change the online submission form on your behalf: 

“The funder provided support in the form of salaries for authors [insert relevant initials], but did not have any additional role in the study design, data collection and analysis, decision to publish, or preparation of the manuscript. The specific roles of these authors are articulated in the ‘author contributions’ section.

Reviewers' comments:

Reviewer's Responses to Questions

**Comments to the Author**

1. Is the manuscript technically sound, and do the data support the conclusions?

Reviewer #1: Yes

Reviewer #2: Yes

Reviewer #3: Yes

2. Has the statistical analysis been performed appropriately and rigorously? 

Reviewer #1: Yes

Reviewer #2: Yes

Reviewer #3: Yes

3. Have the authors made all data underlying the findings in their manuscript fully available?

Reviewer #1: Yes

Reviewer #2: Yes

Reviewer #3: Yes

4. Is the manuscript presented in an intelligible fashion and written in standard English?

Reviewer #1: Yes

Reviewer #2: Yes

Reviewer #3: Yes

5. Review Comments to the Author

Reviewer #1: In the present study the authors have attempted to improve bioavailability of Acetazolamide, a BCS class 4 drug by surface coating using nano- HPA as well as elucidating the possible mechanism for the same. The exact mechanism although still not confirmed by author’s and required further studies which is understandable since absorption being complex and dynamic process with multiple mechanism involved. However, the authors’ have presented a good argument throughout the manuscript with well-structured methodology as well as results and discussions. Thus, I recommend the publication of this manuscript with some minor revisions.

Page 5 line 100- Although coating and preparation method has been cited from earlier studies, it will be good if authors can describe the preparation method of formulation (AZ/HAP) briefly for better understanding for readers.

Page 11 line 248: Reference for figure 1 is missing

Page 20 line 460: As there is no significant difference (p<0.05) between the group, please remove the asterisk mark

Page 25 line 553-554: Discussion regarding no significant difference of solubility between AZ and AZ/HAP formulation in PBS 6.8 is still unclear specially since in case of AZ/HAP formulation being enteric coated. Please elaborate on this for better understanding

Reviewer #2: Article references need to check and update for example Line #139 reference need to add/correct the reference style. Analytical method sensitivity for estimation of drug concentration need to mentioned which used in CaCo2 study.

Reviewer #3: The authors have presented very good findings with experimental data in the manuscript. The authors are appreciated for the way of explanation given in this manuscript. This manuscript can be publishable after minor revisions.

1) Relevance of line number 55-60 with context of present study

2) In line number 72, mention name techniques. Include the following references for the solubility enhancement techniques in line number 72:

a) Diwan, Rimpy, Punna Rao Ravi, Nikita Shantaram Pathare, and Vidushi Aggarwal. "Pharmacodynamic, pharmacokinetic and physical characterization of cilnidipine loaded solid lipid nanoparticles for oral delivery optimized using the principles of design of experiments." Colloids and Surfaces B: Biointerfaces 193 (2020): 111073.

b) Diwan, Rimpy, Punna Rao Ravi, Shubham Ishwar Agarwal, and Vidushi Aggarwal. "Cilnidipine loaded poly (ε-caprolactone) nanoparticles for enhanced oral delivery: optimization using DoE, physical characterization, pharmacokinetic, and pharmacodynamic evaluation." Pharmaceutical Development and Technology 26, no. 3 (2021): 278-290.

c) Diwan, Rimpy, Shareef Khan, and Punna Rao Ravi. "Comparative study of cilnidipine loaded PLGA nanoparticles: process optimization by DoE, physico-chemical characterization and in vivo evaluation." Drug delivery and translational research 10, no. 5 (2020): 1442-1458.

3) Author can mention briefly the preparation of nano-HAP and the source of micron level HAP used to synthesize nano-HAP (line no. 103) ? Did author extract the micron level HAP from teeth or bone in the lab?

4) The author can include the brief procedure of coating of nano-HAP over the AZ particles.

5) Mention the test solution in line 115. Is it phosphate buffer pH 6.8?

6) What is the AZ weight in the 900 mg of AZ/HAP formulation taken for solubility (line 117)?

7) What is the anticipated mechanism of solubility improvement with nano-HAP ?

8) What is the reason of adjusting the AZ concentration to a final concentration of either 540 ppm or 2,240 ppm on mucosal side in Ussing chamber?

9) Did author check the solubility of physical mixture of AZ/HAP?

10) The author should show the error bars in Figure 3A – solubility of AZ alone.

11) What could be the anticipated reason for huge standard deviation in blood concentration of AZ in AZ/HAP formulation at 30 min.?

12) Line 547, author mentioned HAP is known to function as a buffer in a wide pH range. Why AZ solubility is low in DTS 2 although HAP is good buffering agent? What is pKa value of HAP?

13) The authors could have quantified the AZ concentration in the intestine after the oral administration.

14) In authors’ experiment using NPP tissue in the Ussing chamber, at a high level of initial AZ concentration, AZ permeability was significantly higher for the AZ/HAP formulation than for AZ alone, while at a lower level of initial concentration, no significant difference in AZ permeation between the two groups was observed. Why at a high level, AZ permeability is more?

6. PLOS authors have the option to publish the peer review history of their article (what does this mean?). If published, this will include your full peer review and any attached files.

Reviewer #1: No

Reviewer #2: No

Reviewer #3: No

---

## [Author Response · Author response to Decision Letter 0]

31 Mar 2022

Dr. Vivek Gupta,

Academic Editor

PLOS ONE

March 31, 2022

Dear Dr. Gupta,

We are grateful for the valuable comments of the reviewers and the opportunity to revise our manuscript. Our responses to each of the reviewer’s comments are below in red. Please note that page and line references are those in the revised manuscript.

Review Comments

Reviewer #1: In the present study the authors have attempted to improve bioavailability of Acetazolamide, a BCS class 4 drug by surface coating using nano- HAP as well as elucidating the possible mechanism for the same. The exact mechanism although still not confirmed by author’s and required further studies which is understandable since absorption being complex and dynamic process with multiple mechanism involved. However, the authors’ have presented a good argument throughout the manuscript with well-structured methodology as well as results and discussions. Thus, I recommend the publication of this manuscript with some minor revisions.

1) Page 5 line 100- Although coating and preparation method has been cited from earlier studies, it will be good if authors can describe the preparation method of formulation (AZ/HAP) briefly for better understanding for readers.

Author’s response: We agree to this comment. According to your suggestion, we have added a brief description of the method of formulation to the main body. Please refer to the inserted sentence immediately after "... for peroral use [19]" on page 5 lines 105-118.

2) Page 11 line 248: Reference for figure 1 is missing.

Author’s response: Figure 1 was drawn by us on the basis of our Ussing chamber system used in this experiment, and thus there is no reference to be cited. We have added the comment “Authors’ diagram.” (page 12 line 257).

3) Page 20 line 460: As there is no significant difference (p<0.05) between the group, please remove the asterisk mark.

Author’s response: We corrected Table 2 and the asterisk mark was deleted.

4) Page 25 line 553-554: Discussion regarding no significant difference of solubility between AZ and AZ/HAP formulation in PBS 6.8 is still unclear specially since in case of AZ/HAP formulation being enteric coated. Please elaborate on this for better understanding.

Author’s response: The AZ/HAP formulation used in the solubility testing was not enteric coated. In the solubility testing, the solubility of the AZ/HAP formulation in water was higher than that of AZ, but solubility in DTS2 was not significantly different between the two formulations. AZ has been reported to show weak acidity when dissolved in water, while HAP has been reported to have a buffering effect on acidic solutions [(Reference No. 44 in the revised manuscript)]. We therefore considered that this buffering effect contributed to the improvement in solubility of the AZ/HAP formulation in water. On the other hand, in the solubility test using DTS2, since DTS2 itself has a buffering action, we postulated that this buffering action improved the solubility of AZ to the same level as that of the AZ/HAP formulation, and that this is why there was no significant difference in solubility between AZ and the AZ/HAP formulation in the solubility test using DTS2.

Reviewer #2: Article references need to check and update for example Line #139 reference need to add/correct the reference style. Analytical method sensitivity for estimation of drug concentration need to mentioned which used in CaCo2 study.

Author’s response 1: In accordance with your comment, all reference style in our manuscript has been checked and corrected where needed.

Author’s response 2: In the Caco-2 study, HPLC analysis was used to determine drug concentration in the sample from the basolateral tray. Regarding the sensitivity of HPLC analysis, we have provided details of each concentration of the standard drug used for calculating the standard curve, in the section “Pharmacokinetics study” in materials and methods (page 8 lines 172-173). 

Reviewer #3: The authors have presented very good findings with experimental data in the manuscript. The authors are appreciated for the way of explanation given in this manuscript. This manuscript can be publishable after minor revisions.

1) Relevance of line number 55-60 with context of present study.

Author’s response: We intended to show that low solubility development pipelines resulted in decreasing the number of approved new drugs and thus improvement in drug solubility could promote new drug development. To clarify relevance with our present study, the text which Reviewer indicated has been combined with that of the following paragraph and we modified several phrases (See page 3 lines 60-61).

2) In line number 72, mention name techniques. Include the following references for the solubility enhancement techniques in line number 72:

Author’s response: We agree to this comment. We added the proposed references to our reference list and the techniques have been mentioned in the main body (See page 4 lines 72-73).

a) Diwan, Rimpy, Punna Rao Ravi, Nikita Shantaram Pathare, and Vidushi Aggarwal. "Pharmacodynamic, pharmacokinetic and physical characterization of cilnidipine loaded solid lipid nanoparticles for oral delivery optimized using the principles of design of experiments." Colloids and Surfaces B: Biointerfaces 193 (2020): 111073.

b) Diwan, Rimpy, Punna Rao Ravi, Shubham Ishwar Agarwal, and Vidushi Aggarwal. "Cilnidipine loaded poly (ε-caprolactone) nanoparticles for enhanced oral delivery: optimization using DoE, physical characterization, pharmacokinetic, and pharmacodynamic evaluation." Pharmaceutical Development and Technology 26, no. 3 (2021): 278-290.

c) Diwan, Rimpy, Shareef Khan, and Punna Rao Ravi. Comparative study of cilnidipine loaded PLGA nanoparticles: process optimization by DoE, physico-chemical characterization and in vivo evaluation. Drug delivery and translational research 10, no. 5 (2020): 1442-1458.

3) Author can mention briefly the preparation of nano-HAP and the source of micron level HAP used to synthesize nano-HAP (line no. 103) ? Did author extract the micron level HAP from teeth or bone in the lab?

Author’s response: As mentioned above, a brief description of the preparation of nano-HAP and the procedure of coating AZ with nano-HAP has been added to the main body. Please check the sentence beginning with “Briefly” on page 5 lines 105-118.

4) The author can include the brief procedure of coating of nano-HAP over the AZ particles.

Author’s response: As above, a brief description has been added to the in main body, in the text beginning with “Briefly” on page 5 lines 105-118.

5) Mention the test solution in line 115. Is it phosphate buffer pH 6.8?

Author’s response: We used either water or DTS 2 as the test solution for solubility testing. To clarify this, the phrase “the test solution” on lines 124-125 was corrected to “each test solution”.

6) What is the AZ weight in the 900 mg of AZ/HAP formulation taken for solubility (line 117)?

Author’s response: 900 mg of AZ/HAP formulation contains 300mg of AZ. The phrase “(containing 300mg of AZ)” has been inserted immediately after “AZ/HAP” on line 127.

7) What is the anticipated mechanism of solubility improvement with nano-HAP ?

Author’s response: We assume that the mechanism by which nano-HAP improves the solubility of the drug is due to the buffering action of HAP. For more information, please refer to the author's response to Reviewer's comment No.12.

8) What is the reason of adjusting the AZ concentration to a final concentration of either 540 ppm or 2,240 ppm on mucosal side in Ussing chamber?

Author’s response: In the pharmacokinetic study, 3 mL of a solution of AZ or AZ/HAP formulation adjusted to 10 mg/kg in AZ equivalent was orally administered to rats. Since rats with an average body weight of 300 g were used in the experiment, the concentration of 3 mL of the administered solution was about 3,000 ppm in terms of AZ.

Based on the test conditions of the pharmacokinetic study, we investigated whether a pharmaceutical solution with an AZ concentration of 3,000 ppm could be used in the Ussing chamber study. We found that when an AZ / HAP formulation of 2,500 ppm or more was used, the formulation blocked the flow path in the chamber and data collection failed.

On the other hand, the results of the solubility testing revealed that the solubility of the AZ/HAP formulation was about 1,900 ppm at the maximum.

Based on these findings, the drug concentrations used in the Ussing chamber study were set at 2,240 ppm, which exceeds the maximum solubility of both the AZ and AZ/HAP formulations, and 560 ppm, which falls within the solubility range of both formulations.

9) Did author check the solubility of physical mixture of AZ/HAP?

Author’s response: In the present study, we didn’t check the solubility of the physical mixture of AZ and HAP or conduct serial dilution, because in a previous study (described in reference No. 16 (Reference No. 19 in the revised manuscript)) we found that a physical mixture of HAP and the insoluble drug cisplatin showed low solubility compared to that of a mechanically HAP-coated cisplatin formulation.

10) The author should show the error bars in Figure 3A – solubility of AZ alone.

Author’s response: The error bars are actually present but are hidden by plotted data (black circles) because the standard deviation in the AZ group was extremely small. We have added the comment “Error bars not visible in the case of AZ alone as standard deviation was extremely small.” (See page 17 lines 388-389).

11) What could be the anticipated reason for huge standard deviation in blood concentration of AZ in AZ/HAP formulation at 30 min.?

Author’s response: We used enteric-coated AZ/HAP formulation in our pharmacokinetics study, and we believe that there could have been surface variance, leading to a variation in its dissolution in the rat intestine. As described in the author's response to Reviewer #1 comment (1), the enteric-coating procedure was simple, and the quality of the coating such as uniformity and thickness, was not strictly controlled. This may have caused non-uniform enteric coating resulting a variation in dissolution of the coating and of AZ absorption in the rat intestine. We therefore consider that the large SD value for the AZ/HAP group in the pharmacokinetics study may have resulted from non-uniformity in the enteric-coating applied to the AZ/HAP formulation.

12) Line 547, author mentioned HAP is known to function as a buffer in a wide pH range. Why AZ solubility is low in DTS 2 although HAP is good buffering agent? What is pKa value of HAP?

Author’s response: In the solubility testing, the solubility of the AZ/HAP formulation in water was higher than that of AZ, but solubility in DTS2 was not significantly different between the two formulations. AZ has been reported to show weak acidity when dissolved in water, while HAP has been reported to have a buffering effect on acidic solutions. We considered that this buffering effect contributed to the improvement in solubility of the AZ/HAP formulation in water. On the other hand, in the solubility testing using DTS2, since DTS2 itself has a buffering action, we postulated that this buffering action improved the solubility of AZ to the same level as that of the AZ/HAP formulation, and this is why there was no significant difference in solubility between AZ and the AZ/HAP formulation in the solubility testing using DTS2.

 As a result of reexamining Reference No. 40, initially presented by us, we came to the conclusion that it was inappropriate as a reference for explaining the buffering action of HAP as a general theory. Therefore, in the revised manuscript, we have replaced it with the following paper [*] and corrected the phrase "in a wide pH range" in line 559 to "in an acidic pH range". We apologize for our inadequate confirmation of the contents of the reference.

*Reference No. 44 in the revised manuscript: Okazaki M. Chemistry of apatite in teeth and bones, first ed.. Tokai University Press, Tokyo. 1992. 116-119. Japanese.

13) The authors could have quantified the AZ concentration in the intestine after the oral administration.

Author’s response: During our experiments, AZ concentration in the intestine was not recorded. However, we believe its solubility in DTS2 clearly reflects its dissolution in the intestine.

14) In authors’ experiment using NPP tissue in the Ussing chamber, at a high level of initial AZ concentration, AZ permeability was significantly higher for the AZ/HAP formulation than for AZ alone, while at a lower level of initial concentration, no significant difference in AZ permeation between the two groups was observed. Why at a high level, AZ permeability is more?

Author’s response: As postulated in the passage from lines 672-692 in the revised manuscript, nano-HAP particles may interfere with the drug efflux system in intestinal epithelial cells because of the particles’ high affinity for proteins, particularly P-glycoproteins. In addition, it has been reported that nano-HAP particles are internalized in various types of cancer cells via the endocytic pathway, resulting in an anti-cancer effect, in a dose dependent manner, based on inhibition of protein synthesis [*]. Although nano-HAP particles are also taken into normal cells, the amount of nano-HAP internalized is very low compared with that in cancer cells, and it exhibits low cytotoxicity [*]. Considering these findings, we hypothesize that, in the group treated with AZ/HAP formulation at an AZ concentration of 2,240 ppm, the high amount of nano-HAP particles present may have interfered with the function of P-glycoprotein by binding to it, or by inhibiting cellular protein synthesis in normal intestinal epithelial cells.

*Han Y, Li S, Cao X, et al. Different inhibitory effect and mechanism of hydroxyapatite nanoparticles on normal cells and cancer cells in vitro and in vivo. Sci Rep. 2014; 4: 7134. doi: 10.1038/srep07134.

Journal Requirements

1) Please ensure that your manuscript meets PLOS ONE's style requirements, including those for file naming. The PLOS ONE style templates can be found at https://journals.plos.org/plosone/s/file?id=wjVg/PLOSOne_formatting_sample_main_body.pdf and https://journals.plos.org/plosone/s/file?id=ba62/PLOSOne_formatting_sample_title_authors_affiliations.pdf

Author’s response: We have reviewed and adjusted all style in our manuscript and confirmed that it meets the Journal’s requirements.

Author’s response: We have reviewed and corrected the reference list.

3) Thank you for stating the following financial disclosure: 

"This study was funded by Sangi Co., Ltd. The funder provided support in the form of salaries for authors (Kenichi Kaneko and Ryosuke Miyasaka) as well as research funds. The funder had no role in the study design, data collection and analysis, decision to publish, or preparation of the manuscript."

We note that one or more of the authors is affiliated with the funding organization, indicating the funder may have had some role in the design, data collection, analysis or preparation of your manuscript for publication; in other words, the funder played an indirect role through the participation of the co-authors. If the funding organization did not play a role in the study design, data collection and analysis, decision to publish, or preparation of the manuscript and only provided financial support in the form of authors' salaries and/or research materials, please do the following:

a. Review your statements relating to the author contributions, and ensure you have specifically and accurately indicated the role(s) that these authors had in your study. These amendments should be made in the online form.

b. Confirm in your cover letter that you agree with the following statement, and we will change the online submission form on your behalf: 

“The funder provided support in the form of salaries for authors [insert relevant initials], but did not have any additional role in the study design, data collection and analysis, decision to publish, or preparation of the manuscript. The specific roles of these authors are articulated in the ‘author contributions’ section.

Author’s response to a: We have reviewed the author contributions and confirmed the roles of all authors, with only minor adjustment.

Author’s response to b: We added the sentence which you proposed to the cover letter. 

4) In your Data Availability statement, you have not specified where the minimal data set underlying the results described in your manuscript can be found. PLOS defines a study's minimal data set as the underlying data used to reach the conclusions drawn in the manuscript and any additional data required to replicate the reported study findings in their entirety. All PLOS journals require that the minimal data set be made fully available. For more information about our data policy, please see http://journals.plos.org/plosone/s/data-availability.

Author’s response: We have created a supporting information file which contains data underlying our results. Please find attached an excel file entitled “S1 File Minimal data set”.

5) PLOS requires an ORCID iD for the corresponding author in Editorial Manager on papers submitted after December 6th, 2016. Please ensure that you have an ORCID iD and that it is validated in Editorial Manager. To do this, go to ‘Update my Information’ (in the upper left-hand corner of the main menu), and click on the Fetch/Validate link next to the ORCID field. This will take you to the ORCID site and allow you to create a new iD or authenticate a pre-existing iD in Editorial Manager. Please see the following video for instructions on linking an ORCID iD to your Editorial Manager account: https://www.youtube.com/watch?v=_xcclfuvtxQ

Author’s response: At your suggestion, I have already obtained ORCID iD and linked it to my account.

6) We note that you have included the phrase “data not shown” in your manuscript. Unfortunately, this does not meet our data sharing requirements. PLOS does not permit references to inaccessible data. We require that authors provide all relevant data within the paper, Supporting Information files, or in an acceptable, public repository. Please add a citation to support this phrase or upload the data that corresponds with these findings to a stable repository (such as Figshare or Dryad) and provide and URLs, DOIs, or accession numbers that may be used to access these data. Or, if the data are not a core part of the research being presented in your study, we ask that you remove the phrase that refers to these data.

Author’s response: We have created a supporting information file entitled “S1 Fig_Localization of OsteoSense 680EX and Calcein double-labeled nano-HAP particles in rat liver 30 min after oral administration”.

---

## [Decision Letter · Decision Letter 1]

22 Apr 2022

Nano-hydroxyapatite improves intestinal absorption of acetazolamide (BCS Class IV drug) – but how?

PONE-D-21-40400R1

Dear Dr. Kaneko,

We’re pleased to inform you that your manuscript has been judged scientifically suitable for publication and will be formally accepted for publication once it meets all outstanding technical requirements.

Kind regards,

Vivek Gupta

Academic Editor

PLOS ONE

Additional Editor Comments (optional):

Reviewers' comments:

Reviewer's Responses to Questions

**Comments to the Author**

1. If the authors have adequately addressed your comments raised in a previous round of review and you feel that this manuscript is now acceptable for publication, you may indicate that here to bypass the “Comments to the Author” section, enter your conflict of interest statement in the “Confidential to Editor” section, and submit your "Accept" recommendation.

Reviewer #1: All comments have been addressed

Reviewer #2: All comments have been addressed

Reviewer #3: All comments have been addressed

2. Is the manuscript technically sound, and do the data support the conclusions?

Reviewer #1: Yes

Reviewer #2: Yes

Reviewer #3: Yes

3. Has the statistical analysis been performed appropriately and rigorously? 

Reviewer #1: Yes

Reviewer #2: Yes

Reviewer #3: Yes

4. Have the authors made all data underlying the findings in their manuscript fully available?

Reviewer #1: Yes

Reviewer #2: Yes

Reviewer #3: Yes

5. Is the manuscript presented in an intelligible fashion and written in standard English?

Reviewer #1: Yes

Reviewer #2: Yes

Reviewer #3: Yes

6. Review Comments to the Author

Reviewer #1: The authors have addressed all my comments and made all the changes accordingly. So, I recommend acceptance of the manuscript.

Reviewer #2: (No Response)

Reviewer #3: (No Response)

7. PLOS authors have the option to publish the peer review history of their article (what does this mean?). If published, this will include your full peer review and any attached files.

Reviewer #1: No

Reviewer #2: **Yes: **Jayshil Bhatt

Reviewer #3: **Yes: **Himanshu Narendrakumar Bhatt

---

## [Editor Report · Acceptance letter]

12 May 2022

PONE-D-21-40400R1 

Nano-hydroxyapatite improves intestinal absorption of acetazolamide (BCS Class IV drug) – but how? 

Dear Dr. Kaneko:

I'm pleased to inform you that your manuscript has been deemed suitable for publication in PLOS ONE. Congratulations! Your manuscript is now with our production department. 

Kind regards, 

on behalf of

Dr. Vivek Gupta 

Academic Editor

PLOS ONE